# Openness and COVID-19 induced xenophobia: The roles of trade and migration in sustainable development

Leshui He[1], Wen Zhou[2], Ming He[2], Xuanhua Nie[2], Jun He[2]*

1 Department of Economics, Bates College, Lewiston, ME, United States of America, 2 School of Ethnology and Sociology, Yunnan University, Kunming, China

* Jun.he@ynu.edu.cn

**Data Availability Statement:** All relevant data are within the paper and its Supporting Information files.

**Funding:** The research received financial support from the National Social Sciences Foundation of

## Abstract

Along with the plight of the COVID-19 outbreak in 2020 come the xenophobic behaviors and hate crimes against people with Asian descent around the globe. The threat of a public health emergency catalyzed underlying xenophobic sentiments, manifesting them into racial discrimination of various degrees. With most discriminatory acts reported in liberal societies, this article investigates whether an economy more open to trade and migration can be more susceptible to xenophobia. Using our first-hand survey data of 1767 Chinese respondents residing overseas from 65 different countries during February of 2020, we adopt an instrumental variable strategy to identify the causal effect of openness to trade and migration of their residence country on the likelihood of them receiving discriminatory behaviors during the early stage of the COVID-19 outbreak. Our results show that greater openness to trade increases the likelihood of reported xenophobic behaviors, while openness to migration decreases it. On the other hand, stronger trade or immigration relationships with China are associated with less reported discrimination. And these effects primarily influence discriminatory behavior in interpersonal spaces, rather than through media outlets. Our findings highlight nuances of the effect of trade relations on the culture of a society.

## 1 Introduction

Discrimination is a major source of friction and disruption to cooperative relationships across cultural and national boundaries, that are increasingly important in shaping and supporting sustainable development [1]. Our increasingly globalized world sees ever more intense interactions through trade and migration, which, in turn, help to bring down barriers of misunderstanding to mitigate discrimination. On the other hand, globalization also heightens the conflict of interests, breeding hate and xenophobia. In extreme, such sentiments translate into bigotry and even hate crimes that carry extra social harm in addition to their harm to the individual victims [2]. Hate crimes against Asian Americans surged as COVID-19 spreads in the US. More incidents of hate against Asians were brought to light again following incidents in the US in early 2021.

China (No. 16ZDA151) and the Ministry of Education of People's Republic of China (No. project no. 16JJD850015). We also acknowledge all 1767 participants who volunteered to response to this survey.

**Competing interests:** We declare there is no conflict of interests,

Following the news on the COVID outbreak in China since January of 2020, numerous reputable news agencies reported incidents of racial discrimination targeting ethnically Chinese and other Asian residents even *before* the outbreak spreads outside China (He et al. 2020). Such behavior ranges from subtle responses of shunning people of Asian descent to blatant verbal abuses and even physical attacks in public. US major TV network NBC reported on March 26th, 2020, that an online forum, Stop AAPI Hate, received more than 650 direct reports of discrimination against primarily Asian Americans within one week. It appears that the threat of a public health emergency catalyzes underlying xenophobic sentiments, manifesting them into racial discrimination of various degrees.

It is a curious pattern arising from these reports that such shockingly xenophobic behaviors seem to be primarily reported in the most open and liberal economies: Australia, France, Germany, UK, and US. Is it possible that more open societies are more prone to racism in the face of a virus outbreak? Direct observation from the news reports would be misleading due to under-reporting in less liberal societies. To investigate this question, we designed a short survey aimed at ethnically Chinese residents living abroad to collect their observations around the globe.

We are particularly interested in the effect of a country's openness to trade and migration *on* the likelihood of an ethnically Chinese resident observing anti-Asian discriminatory behaviors. Openness to trade and immigration are important components of the globalized world, and we choose to focus on these two aspects to provide a particular perspective regarding the general relationship between globalization and culture. They may mitigate or intensify xenophobic sentiments, and it is an important empirical question to explore the evidence. Openness to trade and migration facilitates more interactions with people outside one's race, ethnicity and nationality, thus potentially improves mutual understanding and cultural exchanges, mitigating xenophobic sentiments [3–5]. In some instances, even low-skilled immigration can improve native unskilled wages, employment, and occupational mobility [6, 7]. They also increase the concentration of immigrant minorities that reduces the likelihood of harassment from the natives by deterring potential threats [8]. On the other hand, openness to trade and migration may threaten interests of native groups on economic grounds [9, 10] or cultural grounds [11], fueling xenophobic sentiments [12–14] and inducing conflicts [15]. It is a curious empirical question to find which effect dominates overall.

Based on an instrumental-variable strategy [16], one literature shows openness to trade and migration has a positive effect on long-run income per capita [17], is associated with more stringent antitrust laws [18], and has little harm on children's health [19]. We closely follow their empirical strategy to address our research question. Our results show that greater openness to trade *increases* the likelihood of reported xenophobic behaviors.

In particular, consider two otherwise similar countries, with one country more open to trade (ranked at the 75th percentile in trade share) than the other (ranked at the 25th percentile in trade share). Our preferred estimates show that a respondent residing in the more open country sees an 80% increase in the likelihood of receiving COVID-19 induced discrimination. By contrast, stronger trade or immigration relationships with China are associated with less reported discrimination.

On the openness to immigration, we found general openness to migration *decreases* reported discrimination. A respondent residing in a country ranked at the 75th percentile of migration share in population (more open to migration) in our sample sees a 38% decrease in the likelihood of receiving COVID-19 induced discrimination compared to those in a country ranked at the 25th percentile. In addition, more immigration from China is associated with a lower level of reported discrimination in our specific context. These findings are in contrast with some previous work in the literature that argues that openness to immigration worsens xenophobic sentiments. An extensive literature at the intersection of psychology, politics and

economics studies the determinants of native residents' attitudes towards immigrants and immigration policies [11, 20]. In addition to the threat to natives' own economic interests through labor market competition [21–23] and the threat of increasing burden of taxation [24, 25], immigration raises sociotropic concerns that induce anti-immigrant attitudes. In particular, anti-immigrant attitudes rise during times of national economic stress [26–29], and of greater anxiety [30]. In contrast to this literature that studies xenophobic *attitudes* of natives as reported by themselves, this article studies xenophobic *behavior* directed at a particular ethnic group that are reported by immigrants.

Our analysis further shows that these effects from trade and migration primarily work through discriminatory behavior observed in interpersonal spaces, rather than through media outlets. Given the increasingly stringent travel restrictions imposed during the pandemic, the world is quickly falling back to a more isolated structure. Re-opening trade and migration would become one of the critical discussions for the post-epidemic world, to which our study contributes a piece of evidence.

The rest of this article proceeds as the following. Section 2 introduces our survey and empirical model, Section 3 describes the data used to estimate our model, and Section 4 presents our main results. Finally, Section 5 discusses the robustness of our results and Section 6 concludes.

## 2 Survey and model specification

The research is conducted as an online survey where participants received the invitation on their cellphones via the mobile app Wechat. We clearly stated that the study is anonymous, and it follows relevant laws and regulations. By accepting the invitation and proceeding to fill the questionnaire and finally submitting their results, the survey subjects proceed to the questionnaire after reading the language describing the survey's intent and purposes. They then filled out the survey questions with the understanding of the study. Our survey is anonymous and did not collect individual identifiers. Before the survey started, the study was approved by IRB at School of Ethnology and Sociology, Yunnan University.

We distributed this short 6-minute online survey through Wechat over the internet, and collected 1767 effective responses from ethnically Chinese residents currently living outside China in 65 different countries. All responses were collected within a window of 7 days from Feb 11th to Feb 17th, 2020—a period when the infection was mostly taking place within mainland China. On Feb 17[th], 2020, WHO reported 70635 confirmed cases in mainland China, and 794 cases outside China (1% of total confirmed cases). In favor of time-sensitivity during the fast-changing outbreak period and wide geographical coverage, we chose not to adopt a prolonged random sampling process. The sample is, therefore, not fully representative, and all results should be interpreted with this caveat. However, we believe our results are reliable, and are complementary to future studies on a similar subject using alternative designs.

During our sampling period, the chance of infection for an average resident in *any* other country remained relatively low and comparable. Therefore, after controlling for officially reported COVID case numbers in each country, relative geographical proximity to China, the risk of contracting the virus from any Asian-looking person is similar across different countries around the world. Thus, controlling for respondent and country characteristics, the variations in the reported discrimination behavior is primarily driven by differences in local characteristics, including openness to trade and migration. Under this conceptual framework, we seek to recover the effect of openness on trade and migration on xenophobic behavior, while addressing potential omitted variable biases using exogenous instruments for openness.

Our measures of coronavirus-related xenophobia come directly from our survey responses. Our questionnaire asked, "Since the virus outbreak in China, have you noticed any related

discriminatory behavior in your working environment and daily life?" The choices include (i) yes; (ii) no; and (iii) not sure. We code an indicator variable "*observed discrimination*" that takes value 1 if the respondent chose "yes" to this question, and value 0 otherwise. We use this variable as our primary measure for xenophobia. If a respondent had answered "yes" or "not sure" in the question above, they were navigated to a multiple-choice question to select the primary type of discrimination that they noticed. Approximately 76% of respondents chose one of three categories: (i) racially discriminatory message against Chinese in the media (29%); (ii) racist rhetoric by native residents against Chinese in public (23%); and (iii) shunning (23%). We code three indicator variables, "racist message in media", "racist rhetoric in public", and "shunning", that takes value 1 if a respondent selected the corresponding category. We use these three indicators as the secondary and measures of xenophobic behavior.

We follow [17] to measure openness to trade and migration. Given any country $c$, we measure the openness to trade by its total value of trade flow (import and export) relative to its national GDP,

$$\text{TSH}_c = \sum_{j \neq c} \text{TSH}_{cj} = \sum_{j \neq c} \frac{\text{Trade}_{cj}}{\text{GDP}_c}$$

where $j$ is an index for all trading partners of $c$. Similarly, the openness to migration is defined by its total immigration flow relative to its population

$$\text{MSH}_c = \sum_{j \neq c} \text{MSH}_{cj} = \sum_{j \neq c} \frac{\text{Immigration}_{cj}}{\text{Population}_c}$$

where $j$ is the origin country of immigration.

The structural equation of our interest is a linear-probability model

$$\text{Discr}_{ict} = \beta_0 + \beta_1 ln\, \text{TSH}_c + \beta_2 ln\, \text{MSH}_c + \beta_3 c_{ct} + \underline{\beta}\, X_{ic} + v_{ic} \qquad (1)$$

where $\text{Discr}_{ict}$ is one of the indicator variables measuring xenophobia reported by respondent $i$ in country $c$ on date $t$. $c_{ct}$ is the number of confirmed COVID patients in country $c$ on date $t$ that controls for the level of threat of the virus outbreak at the local country. And $X_{ic}$ is a vector of individual and country characteristics that include (1) the usual individual demographics: female, age, level of education, occupation, whether the respondent obtained their highest degree of education overseas; and (2) country characteristics: logs of population, area and national GDP, as well as an indicator of whether the country is landlocked. Moreover, because all respondents are Chinese migrants, $X_{ic}$ also includes (3) controls for country $c$'s distance from China, including whether it shares the boarder with China, the log distance from China, and the time difference with China. In addition, to control for heterogeneity in migration history, $X_{ic}$ also includes (4) the respondent's self-reported current migration or visa status (citizen, permanent resident, working visa, student visa, etc.), length of migration, and year of migration. To further control for potentially different discriminatory behavior in crowded public spaces, $X_{ic}$ also includes (5) an indicator of whether the respondent primarily relies on public transportation for their daily commute. Our variables of interest are $\beta_1$ and $\beta_2$, which measure the average marginal effects of a 1% increase in the trade and migration openness, respectively, on the linear probability of a respondent receiving COVID-19 induced discrimination of some form.

However, these key estimates may be biased from the true causal effects if TSH and MSH are endogenous. In particular, model (1) may suffer from an omitted variable bias. For instance, our sampling approach may attract more nationalistic respondents or those with a stronger personal tie to China. If such characteristics are correlated with openness to trade or

migration or their residing country, then our OLS estimates would be biased. If such characteristics are correlated with openness to trade or migration or their residing country, then our OLS estimates would be biased. In addition, some unobservable country-level characteristics, such as a pro-trade and anti-migration ideology, may be correlated with TSH and MSH, while also increasing the likelihood of discrimination, thus leading to a spurious correlation between openness measures and discrimination. For example, [31, 32] argue that a stronger nationalistic sentiment is associated with more xenophobic attitudes.

To address such potential endogeneity concerns, we closely follow the empirical strategy by [17] to construct exogenous instruments for the openness measures that depend entirely on pairwise geographical, language and basic historical relationships. To construct the exogenous instruments, we first estimate the following gravity model at the country-dyad level using pairwise trade and immigration flow data

$$\ln \text{TSH}_{cj} = \gamma_1 \cdot X_c + \gamma_2 \cdot X_j + \gamma_3 \cdot X_{cj} + \epsilon_{cj}, \tag{2}$$

where $X_c$ and $X_j$ are vectors of country characteristics of countries $c$ and $j$, respectively. They each include the country's log population, log area, and the indicator of whether the country is landlocked. $X_{cj}$ is a vector of pairwise characteristics capturing the geographical and cultural relationship between them, including: indicators of whether $c$ and $j$ have shared border, shared language, shared official language, shared time zone, colony history, prior hegemonic relationship, and the interactions of the shared-boarder indicator with log population, log area and landlocked indicator. We then use the estimates from (2) to construct predicted pairwise trade share $\text{PTSH}_{cj}$ and migration share $\text{PMSH}_{cj}$, which depend exclusively on pairwise geographical, language and basic historical relationships. Finally, we aggregate all such pairwise shares over each focal country $c$ to produce $\text{PTSH}_c = \Sigma_{j \neq c} \text{PTSH}_{cj}$ and $\text{PMSH}_c = \Sigma_{j \neq c} \text{PMSH}_{cj}$ as exogenous instrumental variables for $\text{TSH}_c$ and $\text{MSH}_c$, and estimate model (1) with a standard 2-stage least-squares procedure.

Given that our context focuses on discrimination towards the ethnic Chinese, we further decompose the TSH and MSH measures and their instruments into China-related and non-China related measures. Specifically, we isolate out the TSH and MSH with China, and separately construct "leave-one-out" measures and their instruments by excluding trade and migration with China. In particular, it is highly plausible that the instrumental variables for the leave-one-out measures to be exogenous to our survey respondents' unobservable characteristics because they are constructed purely on pairwise geographical relationship between the residence country and third countries that are different from China.

## 3 Data

The key dependent variable and all respondent characteristics come from our survey. Our sample includes a diverse set of respondents. The respondents are 65% female, 55% received their highest degree outside China, 32% routinely use public transportation, 65% below the age of 40 and an median age in the 31–40 year old group. The largest three groups of occupations are professionals (30%), students (27%) and workers (9%). 23% of the sample obtained citizenship in their residing country, 23% obtained permanent residency, and 13% were on working visas. 56% have been living outside China for more than 5 years. 45% started living aboard before 2010. The respondents are highly educated, with 89% with a college education or higher—a much higher level of education than the average Chinese citizen, but not as sizable a deviation among first-generation Chinese migrants of recent generations.

In addition to our survey, we rely on three other data sources for trade, migration and country-level information. The trade share is constructed from the mean trade flows during

**Table 1. Descriptive statistics.**

|  | N | mean | sd | min | max |
|---|---|---|---|---|---|
| *Dependent Variables* received discrimination | 1767 | 0.24 | 0.43 | 0 | 1 |
| racist message in media | 1767 | 0.11 | 0.31 | 0 | 1 |
| racist rhetoric in public | 1767 | 0.084 | 0.28 | 0 | 1 |
| shunning | 1767 | 0.085 | 0.28 | 0 | 1 |
| anti-discrimination advocacy | 1767 | 0.26 | 0.44 | 0 | 1 |
| *Country Characteristics* trade share | 1767 | 0.54 | 0.43 | 0.2 | 2.5 |
| immigration share | 1767 | 0.047 | 0.064 | 0.002 | 1.1 |
| trade with China share | 1767 | 0.061 | 0.062 | 0.008 | 0.5 |
| immigration from China share | 1767 | 0.0031 | 0.0040 | 0 | 0.03 |
| trade (all others) share | 1767 | 0.47 | 0.39 | 0.2 | 2.1 |
| immigration (all others) share | 1767 | 0.044 | 0.062 | 0.002 | 1.1 |
| land locked | 1767 | 0.019 | 0.14 | 0 | 1 |
| population (millions) | 1767 | 123.0 | 146.8 | 0.4 | 1099.0 |
| area (thousands of km$^2$) | 1767 | 4067.8 | 4450.2 | 0.3 | 17243.0 |
| contiguous to China | 1767 | 0.054 | 0.23 | 0 | 1 |
| time difference with China | 1767 | 5.39 | 3.70 | 0 | 12 |
| population-weighted distance to China (km) | 1767 | 7692.0 | 3954.5 | 1168.2 | 18884.5 |
| *Respondent Characteristics* female | 1767 | 0.65 | 0.48 | 0 | 1 |
| highest degree obtained overseas | 1767 | 0.55 | 0.50 | 0 | 1 |
| routine public transport | 1767 | 0.32 | 0.47 | 0 | 1 |

1991 to 2016 from the Correlates of War, Trade 4.0 dataset [33, 34]. To construct our migration share, we use the mean immigration flow from 1991 to 2015 compiled by [35] that is based on data published by the World Bank and United Nations. All country-level characteristics are based on the data compiled by [18]. We use confirmed case data by country published by the World Health Organization (WHO) to control for the status of the virus spread at the country-by-day level. Table 1 reports the descriptive statistics of some key variables. The share of total trade measure may be greater than 1 because the numerator is the sum of total import and export combined, as opposed to net export. One country, United Arab Emirates (UAE), has an immigration flow that is greater than its domestic population. Removing the four responses from UAE does not change our results.

## 4 Results

### 4.1 Main results

Table 2 reports our main results from model (1), with the binary dependent variable being: any discrimination. Column (1) shows that a naive regression on openness to trade and migration alone shows no statistically significant association between reported xenophobia and openness to trade or migration. In column (2), after controlling for country and respondent characteristics, the same relationships remain. Column (3) reports the IV-estimates of the general openness to trade and migration. We find that, once instrumented for the possibly endogenous openness measures with our predicted trade share and migration share, the results are overturned: xenophobia is positively associated with openness to trade, but negatively associated with migration. The Kleibergen-Paap F-statistic for weak identification in the first stage is 10.4, greater than the Stock-Yogo weak ID test critical values at the most stringent 10% maximal IV size of 7.03. We present estimates of the equivalent probit model in Section 5.

**Table 2. Discrimination, openness to trade and immigration.**

| | received discrimination | | | | | |
|---|---|---|---|---|---|---|
| | **(1)** | **(2)** | **(3)** | **(4)** | **(5)** | **(6)** |
| ln(trade share) | 0.00483 (0.0351) | | 0.727*** (0.262) | | | |
| ln(immigration share) | 0.0310 (0.0225) | | -0.354** (0.145) | | | |
| ln(trade (all others) share) | | 0.00891 (0.0340) | | 0.730*** (0.254) | 0.204** (0.0901) | |
| ln(immigration (all others) share) | | 0.0290 (0.0221) | | -0.345** (0.138) | | 0.00373 (0.0496) |
| ln(trade with China share) | | | | | -0.370** (0.144) | |
| ln(immigration from China share) | | | | | | -0.111*** (0.0427) |
| Observations | 1767 | 1767 | 1767 | 1767 | 1767 | 1767 |
| Model | OLS | OLS | IV | IV | IV | IV |
| Kleibergen-Paap F-stat | | | 10.4 | 11.1 | 26.9 | 17.2 |

Robust standard errors in parentheses

* $p < 0.10$

** $p < 0.05$

*** $p < 0.01$

*Notes*: Dependent variable: received discrimination. This table reports regression estimates based on model (1). Column (1) reports the naive OLS regression of the indicator outcome, whether the respondent observed any COVID-related racial discrimination, on two control variables, TSH and MSH. Column (2) reports the OLS estimates of model (1) with the full set of respondent and country controls. Columns (3) through (6) reports the 2SLS estimates of model (1) with different measures of TSH and MSH.

Column (4) presents our preferred estimates. In this regression, we replace the general TSH and MSH measures with the leave-one-out measures that exclude the trade and migration flows with China. The results are very similar to Column (3). Our estimates are both statistically significant and substantive in magnitudes. Specifically, for a one standard deviation increase in the log trade share in GDP, the likelihood of receiving any COVID-19 induced discrimination increases by 50% (0.73 × 0.68) or 1.1 standard deviations (0.73 × 0.68 / 0.44).

Put in a slightly different way, a respondent in the country ranked at the 75th percentile of trade share in GDP

in our sample sees an 80% increase in the likelihood of receiving COVID-19 induced discrimination compared to an otherwise identical country ranked at the 25th percentile at openness to trade. On the contrary, with a one standard deviation increase in the log migration share in population, the likelihood of receiving any COVID-19 induced discrimination decreases by 38% (-0.35 × 1.09) or 0.87 standard deviations (-0.35 × 1.09 / 0.44).

## 4.2 China-related trade and immigration

We further explore whether trade and migration with China have a different impact on xenophobic attitudes towards Chinese migrants, comparing to general openness to trade and migration. We analyze this relationship by studying the openness to trade and migration separately and decomposing the general openness into openness to trade to China and to all other countries. We treat both the shares with and excluding China as endogenous variables, and construct IVs for each variable separately. The results are presented in Table 2, columns (5) and (6). The first-stage estimates are presented in Table 3.

We make two observations from this analysis. First, after controlling for the openness to trade and migration with China, the effects of general openness to trade and migration are largely robust. Second, we found that, in contrast to the general openness to trade, more trade directly with China and more migration from China each contributes significantly to a lower level of xenophobic behavior in this context.

Specifically, a one standard deviation increase in the log trade share with China in GDP reduces the likelihood of receiving any COVID-19 induced discrimination by 28% (-0.37 × 0.76) or 0.6 standard deviations (-0.37 × 0.76 / 0.44). And a one standard deviation increase in the log migration share from China in population lowers dependent variable by 36% (0.11 × 3.3) or 0.82 standard deviations (0.11 × 3.3 / 0.44). In percentile terms of openness to trade (migration) with China, a respondent in the 75th-percentile country, on average, have a 35% (21%) lower likelihood of receiving COVID-19 induced discrimination comparing to an otherwise identical country ranked at 25th percentile on this measure. These results are consistent with general intuition that increased *direct* social and economic interactions reduce xenophobia.

## 4.3 Types of discrimination

Table 4 reports the IV estimates of our preferred model with our secondary dependent variables: racist rhetoric in public, shunning, and racist message in media. The results provide additional nuances behind our main result, but of another dimension. Columns (1) and (2) show that, as is consistent with our main result, general openness to trade (migration) is likely to increase (decrease) xenophobic behavior in personal spaces in the form of racist public comments or shunning. However, column (3) shows that openness to trade or migration has no effect on public media. These mixed effects suggest that the effects of trade and migration on reported xenophobic behavior are primarily driven by actions in interpersonal space, as opposed to through the public media.

## 5 Robustness

Our findings are robust to a number of alternative analyses. In this section, we (1) interrogate the model specification by investigating potentially omitted variables; (2) use probit and IV

**Table 3. Discrimination, openness to trade and immigration: 1st-stage result.**

| | (1) | (2) | (3) | (4) | (5) | (6) | (7) | (8) |
|---|---|---|---|---|---|---|---|---|
| | ln(trade share) | ln(immigration share) | ln(trade (all others) share) | ln(immigration (all others) share) | ln(trade (all others) share) | ln(trade with China share) | ln(immigration (all others) share) | ln(immigration from China share) |
| ln predicted share—total trade | 1.042*** (0.0598) | 2.324*** (0.0829) | | | | | | |
| ln predicted share—total immigration | 0.341*** (0.0355) | 0.242*** (0.0445) | | | | | | |
| ln predicted share—trade (all others) | | | 0.951*** (0.0569) | 2.184*** (0.0730) | 1.164*** (0.0506) | 0.432*** (0.0564) | | |
| ln predicted share—immigration (all others) | | | 0.340*** (0.0341) | 0.263*** (0.0406) | | | 0.935*** (0.0546) | -1.229*** (0.194) |
| ln predicted share—trade with China | | | | | 0.635*** (0.0965) | 0.825*** (0.0867) | | |
| ln predicted share—immigration from China | | | | | | | 1.155*** (0.0546) | 0.289 (0.258) |
| Observations | 1767 | 1767 | 1767 | 1767 | 1767 | 1767 | 1767 | 1767 |

Robust standard errors in parentheses

* $p < 0.10$

** $p < 0.05$

*** $p < 0.01$

Notes: This table reports 1st stage estimates of the constructed instruments associated with Table 2. The Kleibergen-Paap F-statistics of the first-stage are reported in Table 2.

**Table 4. Received discrimination by type.**

| | (1) | (2) | (3) |
|---|---|---|---|
| | racist rhetoric in public | shunning | racist message in media |
| ln(trade (all others) share) | 0.659*** (0.216) | 0.296** (0.147) | -0.0659 (0.115) |
| ln(immigration (all others) share) | -0.375*** (0.117) | -0.137* (0.0799) | 0.0768 (0.0626) |
| Observations | 1767 | 1767 | 1767 |
| Model | IV | IV | IV |
| Kleibergen-Paap F-stat | 11.1 | 11.1 | 11.1 |

Robust standard errors in parentheses

* $p < 0.10$

** $p < 0.05$

*** $p < 0.01$

*Notes*: This table reports 2SLS estimates of model (1) using secondary measures of xenophobia as dependent variables.

probit models as alternatives to the linear probability models; (3) address concerns about our sample representativeness using Wave 6 data of the World Value Survey, in which two questions asked about respondents' attitude toward other races and migrants; and (4) use alternative trade and migration data sources to construct openness to trade and migration.

One concern with our specification of model (1) is that more open societies may see more varieties of speech and opinions. Thus the relationship between observed discrimination and openness to trade and migration may be spurious—they are merely reflecting their effects on the diversity of speech, as opposed to xenophobic sentiment per se. Under this hypothesis, our estimated effects of openness to trade and migration would be mitigated once we control for some measure of the opposite speech. To investigate this possibility, we augment our baseline model by including respondents' reports of anti-discrimination advocacy as the control of speech on the other extreme of the spectrum. Table 5 columns (1)-(3) reports the IV estimates. Contrary to the above hypothesis, the effects from openness to trade and migration are hardly mitigated compared to our baseline results in Table 2.

A second concern is that local responses may drive the reported discriminatory responses to the virus outbreak, such as information provided on the outbreak, as opposed to the underlying xenophobia. If governments and employers in more open economies adopt more aggressive measures on travel restrictions, travel history tracking, or contact tracing, then such measures may prime native residents to behave more aggressively toward those of Asian descent. This hypothesis implies that controlling for local preventative measures to the virus would mitigate our key effects. To check this, we use information collected in our survey regarding the type of preventative measured adopted by their employer and local government as additional controls. Specifically, from a list of preventative measures, the respondent was asked to check all those applied to their situation. The options include (1) requiring travel history from those recently traveled to China; (2) requiring travel history from those of Chinese descent; (3) requiring self-isolation from those recently traveled to China; (4) requiring self-isolation from those of Chinese descent; and (5) requesting a reduction of business interactions with China. We create a list of indicator variables for each preventative measure and add them to our regression. Table 5 column (4)-(6) report the results. Our main results remain robust to the inclusion of controls for these preventative measures. Thus there is no evidence that our results are driven by priming from local policy responses.

A third concern regards our linear probability specification of the model. We favor the linear probability model for transparency and ease of interpretation, but the model could be mis-

**Table 5. Mitigating factors: Anti-discrimination advocacy, measures of disease control and prevention.**

| | received discrimination | | | | | |
|---|---|---|---|---|---|---|
| | **(1)** | **(2)** | **(3)** | **(4)** | **(5)** | **(6)** |
| ln(trade (all others) share) | 0.707*** (0.246) | 0.206** (0.0898) | | 0.714*** (0.247) | 0.196** (0.0909) | |
| ln(trade with China share) | | -0.370** (0.144) | | | -0.368** (0.147) | |
| ln(immigration (all others) share) | -0.332** (0.134) | | 0.00318 (0.0492) | -0.340** (0.134) | | -0.00712 (0.0498) |
| ln(immigration from China share) | | | -0.109*** (0.0423) | | | -0.116*** (0.0435) |
| anti-discrimination advocacy | 0.119*** (0.0280) | 0.0982*** (0.0244) | 0.114*** (0.0289) | 0.108*** (0.0284) | 0.0866*** (0.0249) | 0.0982*** (0.0292) |
| prev.: require travel history to China | | | | -0.00320 (0.0272) | 0.0166 (0.0249) | 0.0131 (0.0285) |
| prev.: require travel history from Chinese | | | | 0.0646*(0.0381) | 0.0697** (0.0347) | 0.0568 (0.0447) |
| prev.: require isolation for traveler from China | | | | 0.0329 (0.0254) | 0.0177 (0.0217) | 0.0635** (0.0309) |
| prev.: require isolation for Chinese | | | | 0.0201 (0.0296) | 0.0149 (0.0260) | 0.00503 (0.0318) |
| prev.: reduce interactions with China | | | | 0.0449 (0.0311) | 0.0466* (0.0274) | 0.0234 (0.0372) |
| Observations | 1767 | 1767 | 1767 | 1767 | 1767 | 1767 |
| Model | IV | IV | IV | IV | IV | IV |
| Kleibergen-Paap F-stat | 11.3 | 26.9 | 17.3 | 11.3 | 25.7 | 16.5 |

Robust standard errors in parentheses

* $p < 0.10$

** $p < 0.05$

*** $p < 0.01$

*Notes*: This table reports IV regression estimates based on a variation of model (1). Columns (1)-(3) add the indicator of whether the respondent observed any anti-discrimination advocacy in the media or from the residing country's government; columns (4)-(6) add a set of indicator variables of local disease control and prevention measures.

specified. To check this possibility, we estimate probit and IV probit counterparts of model 1. Tables 6 and 7 report the probit estimates as counterparts to Tables 2 and 4, and our key results remain robust.

A fourth concern relates to our sampling framework and the specific context we study. Because our response data is drawn from a non-random sampling framework, we would like to compare results with an alternative data source with individual-level cross-country

**Table 6. Discrimination, openness to trade and immigration (probit).**

| | received discrimination | | | | | |
|---|---|---|---|---|---|---|
| | **(1)** | **(2)** | **(3)** | **(4)** | **(5)** | **(6)** |
| ln(trade share) | 0.0237 (0.108) | | 2.379*** (0.684) | | | |
| ln(immigration share) | 0.114 (0.0776) | | -1.143*** (0.379) | | | |
| ln(trade (all others) share) | | 0.0368 (0.105) | | 2.390*** (0.670) | 0.834** (0.343) | |
| ln(immigration (all others) share) | | 0.107 (0.0762) | | -1.110*** (0.367) | | 0.0145 (0.200) |
| ln(trade with China share) | | | | | -1.547** (0.633) | |
| ln(immigration from China share) | | | | | | -0.375** (0.161) |
| Observations | 1767 | 1767 | 1767 | 1767 | 1767 | 1767 |
| Model | Probit | Probit | IV Probit | IV Probit | IV Probit | IV Probit |

Robust standard errors in parentheses

* $p < 0.10$

** $p < 0.05$

*** $p < 0.01$

*Notes*: This table reports results of the probit and IV-probit estimation that is equivalent to model (1). The table layout corresponds to that of Table 2.

**Table 7. Received discrimination by type (probit).**

| | (1) | (2) | (3) |
|---|---|---|---|
| | racist rhetoric in public | shunning | racist message in media |
| ln(trade (all others) share) | 3.255*** (0.882) | 1.948** (0.836) | -0.666 (0.822) |
| ln(immigration (all others) share) | -1.872*** (0.487) | -0.937** (0.454) | 0.657 (0.453) |
| Observations | 1756 | 1763 | 1763 |
| Model | IV Probit | IV Probit | IV Probit |

Two-step standard errors in parentheses

* $p < 0.10$

** $p < 0.05$

*** $p < 0.01$

*Notes*: This table reports estimates of the probit models that is equivalent to model (1), using secondary measures of xenophobia as dependent variables. The model in each column is the counterpart of column (4) in Table 6.

information on general xenophobic measures. The best approximation we find is the Wave 6 data of the World Value Survey (WVS), which covers 55 countries with surveys conducted from 2010–2014. Their questionnaire asks, "On this list are various groups of people. Could you please mention any that you would not like to have as neighbors?" and two of the answers that approximate for a measure of xenophobia are: (1) people of a different race; and (2) immigrants/foreign workers. We use indicator variables for those who selected these options as dependent variables in a counterpart of model (1), while controlling for all the demographics information available in the WVS dataset. Tables 8 and 9 report the results. For our preferred model, reported in column (4), the Kleibergen-Paap F-statistic for weak identification in the first stage is 296.2, greater than the Stock-Yogo weak ID test critical values at the most stringent 10% maximal IV size of 7.03. The first-stage estimates shows that instruments are strong, we omit them here for brevity and the results are available upon request. These estimates

**Table 8. Discrimination, openness to trade and immigration (WV6).**

| | Dislike immigrant/foreigner as neighbor | | | | | |
|---|---|---|---|---|---|---|
| | (1) | (2) | (3) | (4) | (5) | (6) |
| ln(trade share) | 0.160*** (0.00433) | | 1.436*** (0.309) | | | |
| ln(immigration share) | 0.0513*** (0.00205) | | -0.181** (0.0704) | | | |
| ln(trade (all others) share) | | 0.151*** (0.00429) | | 0.0308 (0.0509) | 0.519*** (0.0135) | |
| ln(immigration (all others) share) | | 0.0480*** (0.00201) | | 0.101*** (0.00959) | | 0.644*** (0.0282) |
| ln(trade with China share) | | | | | -0.0256*** (0.00534) | |
| ln(immigration from China share) | | | | | | 0.138*** (0.00639) |
| Observations | 82527 | 82527 | 82527 | 82527 | 83765 | 82527 |
| Model | OLS | OLS | IV | IV | IV | IV |
| Kleibergen-Paap F-stat | | | 15.2 | 296.2 | 4720.8 | 301.8 |

Robust standard errors in parentheses

* $p < 0.10$

** $p < 0.05$

*** $p < 0.01$

*Notes*: This table reports regression estimates based on model (1) using the Wave 6 (2010–2012) data of the World Value Survey. The dependent variable is an indicator variable where the respondent mentioned that they did not want *immigrants or foreign workers* as their neighbor (based on question V39). The table layout is equivalent to that of Table 2.

**Table 9. Discrimination, openness to trade and immigration (WV6).**

| | Dislike different race as neighbor | | | | | |
|---|---|---|---|---|---|---|
| | **(1)** | **(2)** | **(3)** | **(4)** | **(5)** | **(6)** |
| ln(trade share) | 0.192*** (0.00449) | | 1.274*** (0.292) | | | |
| ln(immigration share) | 0.0643*** (0.00215) | | -0.112* (0.0667) | | | |
| ln(trade (all others) share) | | 0.182*** (0.00447) | | 0.203*** (0.0528) | 0.636*** (0.0145) | |
| ln(immigration (all others) share) | | 0.0607*** (0.00211) | | 0.0947*** (0.0100) | | 0.781*** (0.0333) |
| ln(trade with China share) | | | | | -0.0527*** (0.00571) | |
| ln(immigration from China share) | | | | | | 0.168*** (0.00749) |
| Observations | 82527 | 82527 | 82527 | 82527 | 83765 | 82527 |
| Model | OLS | OLS | IV | IV | IV | IV |
| Kleibergen-Paap F-stat | | | 15.2 | 296.2 | 4720.8 | 301.8 |

Robust standard errors in parentheses

* $p < 0.10$

** $p < 0.05$

*** $p < 0.01$

*Notes*: This table reports regression estimates based on model (1) using the Wave 6 (2010–2012) data of the World Value Survey. The dependent variable is an indicator variable where the respondent mentioned that they did not want *someone of a different race* as their neighbor (based on question V37). The table layout is equivalent to that of Table 2.

confirm our key finding that openness to trade increases the likelihood of xenophobia, but yield different results on the effect of openness to migration. It is important to highlight that a close comparison of these results with ours should be interpreted with caution, given a number of differences between these two sets of surveys. (1) The question from WVS is hypothetical, while ours asked for observed behavior in recent history; (2) the respondents of WVS are those potentially display discriminatory behaviors toward others, while ours are potential victims to such behaviors; and (3) the question in WVS asks about the underlying willingness having someone as a neighbor, which is a starkly different context from acting aggressively toward others under the threat of a virus outbreak. These two sets of data, therefore, may be measuring different underlying phenomenon and are subject to different behavioral biases [36]. Overall, our interpretation of these mixed results is that there seemed to have a substantive and robust effect between openness to trade and migration, despite possible discrepancies in specific results. Above all, our findings call for further research in alternative scenarios, sampling framework designs, and empirical methods.

Finally, we investigate whether our results are sensitive to alternative trade and immigration data sources. For alternative trade flow data, we use the Direction of Trade Statistics by IMF from 1991 to 2015. And for the migration flow data, we use the mean immigration flow during 1991 to 2000 in the World Bank Global Bilateral Migration Database [37] to construct our migration share. The results are reported in Tables 10 and 11, and our main results remain largely robust.

## 6 Concluding remarks

This paper analyzes a cross-country survey data during the early stage of the COVID-19 outbreak to study the effect of openness to trade and migration on the likelihood of xenophobic and racially discriminatory behavior. Our results show that greater openness to trade increases the likelihood of reported xenophobic behaviors, while openness to migration decreasing it. Our findings highlight previously unknown benefit for policies that foster openness to

**Table 10. Discrimination, openness to trade and immigration (alternative trade data from IMF).**

| | received discrimination | | | | | |
| | **(1)** | **(2)** | **(3)** | **(4)** | **(5)** | **(6)** |
|---|---|---|---|---|---|---|
| ln(trade share) | 0.0312 (0.0298) | | 0.522*** (0.202) | | | |
| ln(immigration share) | -0.00284 (0.0181) | | -0.248** (0.113) | | | |
| ln(trade (all others) share) | | 0.0305 (0.0285) | | 0.406*** (0.149) | -5.071 (33.47) | |
| ln(immigration (all others) share) | | -0.00128 (0.0174) | | -0.182** (0.0845) | | -0.0694 (1.031) |
| ln(trade with China share) | | | | | 7.198 (47.62) | |
| ln(immigration from China share) | | | | | | 3.591 (26.51) |
| Observations | 1764 | 1764 | 1764 | 1764 | 1764 | 1767 |
| Model | OLS | OLS | IV | IV | IV | IV |
| Kleibergen-Paap F-stat | | | 15.8 | 25.2 | 0.0 | 0.0 |

Robust standard errors in parentheses

* $p < 0.10$

** $p < 0.05$

*** $p < 0.01$

*Notes*: This table reports regression estimates based on model (1), using alternative data sources for bilateral trade and migration data. The table layout is equivalent to that of Table 2.

migration. Interestingly, we found that trade and migration with China mitigate reported xenophobia, and the effects of trade and migration on discrimination primarily manifest through actions in interpersonal spaces. These findings shed new light on the nuances of the interactions between economic, social and cultural interactions.

**Table 11. Discrimination, openness to trade and immigration (alternative trade and migration data): 1st-stage result.**

| | **(1)** | **(2)** | **(3)** | **(4)** | **(5)** | **(6)** | **(7)** | **(8)** |
| | ln(trade share) | ln(immigration share) | ln(trade (all others) share) | ln(immigration (all others) share) | ln(trade (all others) share) | ln(trade with China share) | ln(immigration (all others) share) | ln(immigration from China share) |
|---|---|---|---|---|---|---|---|---|
| ln predicted share—total trade | 0.553*** (0.0552) | 1.439*** (0.0842) | | | | | | |
| ln predicted share—total immigration | 0.527*** (0.0492) | 0.588*** (0.0697) | | | | | | |
| ln predicted share—trade (all others) | | | 0.518*** (0.0531) | 1.458*** (0.0739) | 0.742*** (0.0517) | 0.527*** (0.0491) | | |
| ln predicted share—immigration (all others) | | | 0.568*** (0.0462) | 0.574*** (0.0664) | | | 0.973*** (0.0681) | 0.0506 (0.236) |
| ln predicted share—trade with China | | | | | 0.707*** (0.160) | 0.471*** (0.141) | | |
| ln predicted share—immigration from China | | | | | | | 1.301*** (0.0769) | 0.00445 (0.350) |
| Observations | 1764 | 1764 | 1764 | 1764 | 1764 | 1764 | 1767 | 1767 |

Robust standard errors in parentheses

* $p < 0.10$

** $p < 0.05$

*** $p < 0.01$

Notes: This table reports 1st stage estimates of the constructed instruments associated with Table 10. The Kleibergen-Paap F-statistics of the first-stage are reported in Table 10.

In an attempt to reach an extensive coverage across the globe in a fast-changing environment, our sampling framework is not fully representative. Yet we believe this article raises a critical issue in the discussions on the social impacts of COVID-19, and it calls for additional work on this topic to provide further evidence.

## Supporting information

**S1 Questionnaire.**
(PDF)

**S1 Data.**
(DTA)

## Author Contributions

**Conceptualization:** Leshui He, Ming He, Jun He.

**Data curation:** Leshui He, Wen Zhou, Xuanhua Nie, Jun He.

**Formal analysis:** Leshui He, Wen Zhou, Jun He.

**Funding acquisition:** Ming He, Jun He.

**Investigation:** Leshui He, Wen Zhou, Ming He, Xuanhua Nie, Jun He.

**Methodology:** Leshui He, Wen Zhou, Ming He, Jun He.

**Project administration:** Jun He.

**Resources:** Leshui He, Wen Zhou.

**Supervision:** Ming He.

**Visualization:** Leshui He.

**Writing – original draft:** Leshui He, Jun He.

**Writing – review & editing:** Leshui He, Wen Zhou, Ming He, Xuanhua Nie, Jun He.

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
