## [Decision Letter · Decision Letter 0]

23 Dec 2020

PONE-D-20-32641

Openness and COVID-19 induced xenophobia: 

The roles of trade and migration in sustainable development

PLOS ONE

Dear Dr. He,

Thank you for submitting your manuscript to PLOS ONE. After careful consideration, we feel that it has merit but does not fully meet PLOS ONE’s publication criteria as it currently stands. Therefore, we invite you to submit a revised version of the manuscript that addresses the points raised during the review process.

We look forward to receiving your revised manuscript.

Kind regards,

Shang E. Ha, Ph.D.

Academic Editor

PLOS ONE

Journal Requirements:

3. Thank you for stating the following in your Competing Interests section: 'no'

a. Please complete your Competing Interests statement to state any Competing Interests. If you have no competing interests, please state "The authors have declared that no competing interests exist.", as detailed online in our guide for authors at http://journals.plos.org/plosone/s/submit-now

4. We note that Figure 1 in your submission contains map images which may be copyrighted.

We require you to either (a) present written permission from the copyright holder to publish this figure specifically under the CC BY 4.0 license, or (b) remove the figure from your submission:

b. If you are unable to obtain permission from the original copyright holder to publish these figure under the CC BY 4.0 license or if the copyright holder’s requirements are incompatible with the CC BY 4.0 license, please either i) remove the figure or ii) supply a replacement figure that complies with the CC BY 4.0 license. Please check copyright information on all replacement figures and update the figure caption with source information. If applicable, please specify in the figure caption text when a figure is similar but not identical to the original image and is therefore for illustrative purposes only.

Reviewers' comments:

Reviewer's Responses to Questions

**Comments to the Author**

1. Is the manuscript technically sound, and do the data support the conclusions?

Reviewer #1: Partly

Reviewer #2: Partly

2. Has the statistical analysis been performed appropriately and rigorously? 

Reviewer #1: Yes

Reviewer #2: Yes

3. Have the authors made all data underlying the findings in their manuscript fully available?

Reviewer #1: Yes

Reviewer #2: No

4. Is the manuscript presented in an intelligible fashion and written in standard English?

Reviewer #1: Yes

Reviewer #2: Yes

5. Review Comments to the Author

Reviewer #1: This paper investigates whether more open economy in terms of trade and migration has an impact on observed discriminatory behaviors empirically. The theoretical frameworks build on existing research demonstrating that openness to trade and immigration facilitates more interactions with people outside one’s race, ethnicity and nationality, thus potentially improves mutual understanding and cultural exchanges and mitigating xenophobic sentiments. Using survey data on ethnically Chinese residents’ xenophobic experiences living abroad at the early stage of the pandemic in February, 2020, the authors find that greater openness to trade decreases the likelihood of reported xenophobic behaviors while openness to migration increases it.

I think that the paper will be interesting to scholars in political economy and public opinion. The paper addresses an interesting but unexplored question. I recommend revisions before the paper appears in Polis-one or another journal.

1. Theory

The objective of this paper is to examine whether there is a causal effect of trade and immigration policy on discriminatory behaviors observed by ethnically Chinese residents living abroad at the early time of the covid-19 pandemic. Although the paper is empirical-oriented one, we still need some theoretical explanations of how changes in trade and immigration affect discriminatory behaviors observed. In this sense, the paper has some issues to be addressed. First, it does not explain why we focus on the effects of immigration and trade on discriminatory behaviors observed. Globalization can have various dimensions such as trade, foreign direct investment, immigration, and capital market liberalization. Given the various aspects of globalization, I am wondering why the authors pay particular attention to the two dimensions of globalization – trade and immigration, not trade and FDI or trade capital market liberalization, etc. The authors would justify this point well in the main text.

Second, I am wondering how trade and immigration influence discriminatory behaviors observed among ethnically Chinese residents living abroad. I think that a large body of literature in international political economy has explained underlying mechanisms linking trade and immigration to xenophobic behaviors. Yet the authors do not present these mechanisms in a systematic way, while just describing some selective studies supporting their theoretical frameworks. In doing so, the authors do not describe whether trade and immigration have a negative, positive, or null impact on xenophobic behavior observed. The authors would elaborate some testable hypotheses on the effect of trade and immigration on xenophobic behavior observed.

2. Empirics

The authors predict that an increase in trade and immigration can have a negative, positive, or null effect on xenophobic behaviors observed. The questionnaire is “Since the virus outbreak in China, have you noticed any related discriminatory behavior in your working environment and daily life?.” The underlying logic behind the hypothesis is that openness to trade and immigration facilitates more interactions with people outside one’s race, ethnicity and nationality, thus potentially improves mutual understanding and cultural exchanges and mitigating xenophobic sentiments. Given this logic, how can we know that trade and immigration affect local natives in a way that the authors expect? To figure out the mechanisms, wouldn’t it be better to do some survey experiments to local natives rather than ethnically Chinese residents? Simply asking ethnically Chinese residents living abroad does not show that they identify the causal mechanisms between trade and immigration and xenophobic behaviors as local natives are main actors under their theoretical framework.

Related to the above point, another concern is that the authors use trade and immigration indicators at the national level. Employing the national-level indicators capturing trade and immigration does not tell that individuals are affected by them. Although countries may be more economically opened in terms of trade and immigration, it does not necessarily mean that individuals know, experience, and perceive them objectively and subjectively. If the authors attempt to uncover the causal mechanisms, it would be better to use some survey questionnaires to measure how individual respondents know, experience, and perceive the degree of trade and immigration at the individual level.

Reviewer #2: This paper traces the country-specific factors that contribute to anti-Chinese discrimination in other countries under the shadow of a pandemic first spread out in China. There are many merits in this paper. Methodologically, this study is carefully done following Ortega and Peri (2014)'s IV approach using the dyadic bilateral geographical and cultural distance. The topic is very timely and critical as the pandemic has been prevailing around the world. Nonetheless, I have several points of reservation and concern regarding this research.

1. It is entirely in a black box how the respondents are recruited. Even a snowballing sampling, the readers need to know how the respondents are sampled, and who they are, how the survey was done in which language. For instance, it might be possible that these respondents were recruited from the more concerned or more nationalistic population of Chinese immigrants. The platform used in recruitment (WeChat) makes this more likely. Also, it appears that there is no compensation for the survey to the respondents. If you are not paid, what would have been the motivation for the survey-takers to participate in this survey, other than they are particularly concerned about the anti-Chinese atmosphere or feel patriotic about the difficulties their home country was going through? I do not think the WVS analysis remedies this issue.

2. Conceptually, I was not entirely clear whether the paper is about anti-immigrants, anti-China, anti-Chinese, or anti-Asian, or just broad xenophobia: all these have different implications for hypothesizing and analyses. The timing of the survey was 7 days from Feb 11th to Feb 17th, 2020, which the authors described it was a period when the infection was mostly taking place within mainland China. However, the epidemic already took place in South Korea and Japan (in the cruise ship) on a massive scale and in Taiwan and Hong Kong as well. I think this makes Anti-Asian sentiment a better angle, but at least I hope this issue can be discussed and clarified at the beginning.

3. Empirically, the primary issues I had were related to how to adopt Ortega and Peri's approach. I think this paper's setup is rather China- or East Asia- specific and different from the general perspective in Ortega and Peri. So the right approach would be to take the share of trade with "China" and the share of "Chinese immigrants" in the population, rather than general trade or immigration. Many European countries probably have many immigrants from neighboring European states, but a few from China. I could not think of why and how this would matter in the same way as, say, in South Korea, where a large proportion of immigrants must be from China.

4. Second, I am very concerned about the correlation between trade and immigration and the fact that the authors use these variables together in all models. I suspect this might drive the results of the paper. First, the authors use the same IV for both trade and immigration. Second, as shown in Figure 1, the vast majority of responses came from the US, Australia, and Canada: all of them are high immigration and high trade countries. Third, Ortega and Peri (2014) use the two variables separately and together, which I believe this paper should do. Also, the authors need to report the first stage of 2SLS, at least in the appendix.

5. Finally, because all survey was taken after the outbreak of the epidemic and the question was specifically about "since the outbreak," I was not sure if the discrimination got worse than or the same as before. Especially, due the deteriorating relationship between China and the US along with some western countries since at least 2018, maybe the discrimination was rising even before the pandemic.

6. I have questions about the data sources: Why trade data (2012-2016) is from CoW, not the World Bank or the WTO? Why are migration data from 1991-2000? There was almost no migration from China to Africa back then?

7. Some minor points:

a. P.6. says "approximately 76% of respondents chose one of three categories: (i) racially discriminatory message against Chinese in the media (29%); (ii) racist rhetoric by native residents against Chinese in public (23%); and (iii) shunning (23%)." 76% means these categories are exclusive to each other? What if one respondent experienced many of these?

b. On P.4 the authors provide some key descriptive statistics, which is very confusing. "In particular, our preferred estimates show that a respondent residing in a country ranked at the 75th percentile of trade share in GDP in our sample sees a 43% decrease in the likelihood of receiving COVID-19 induced discrimination compared to those in a country ranked at the 25th percentile. In contrast, a respondent in the 75th-percentile country in the share of immigrants in population, on average, has an 86% greater likelihood of receiving COVID-19 induced discrimination compared to those in the counterpart ranked at the 25th percentile." It was unclear because I could not figure out whether the percentile was ascending or descending order. It was only evident after I saw the empirical results in later pages.

c. Isn't it also possible that observed discrimination can be subject to the fear that the respondents at the time?

d. Table1: how about age? Why public transportation?

e. Local media on P. 10 mean domestic media?

f. Can the authors provide full survey questions and an screenshot of the survey in the appendix?

6. PLOS authors have the option to publish the peer review history of their article (what does this mean?). If published, this will include your full peer review and any attached files.

Reviewer #1: No

Reviewer #2: No

---

## [Author Response · Author response to Decision Letter 0]

17 Feb 2021

Response to reviewer comments

We thank the editor and the two reviewers for their detailed and insightful comments. 

First and foremost, we’d like to note that the result in this revision has changed dramatically from the previously submitted version. However, our research question and empirical model remain the same. 

Two main reasons led to the change in results. First, in the process of revising the submitted manuscript, we identified a coding error in the data processing process, the correction of the error led to major changes in the result. In addition, based on the suggestion of the reviewer 2, we collected more recent bilateral immigration data and re-estimated our model. The updated data also led to changes in the result. Despite the changes, we are more confident in the results presented in this revised version.

In the remaining part of this note, we offer a more detailed response to the reviewers’ comments.

Reviewer 1:

1. Theory

The objective of this paper is to examine whether there is a causal effect of trade and immigration policy on discriminatory behaviors observed by ethnically Chinese residents living abroad at the early time of the covid-19 pandemic. Although the paper is empirical-oriented one, we still need some theoretical explanations of how changes in trade and immigration affect discriminatory behaviors observed. In this sense, the paper has some issues to be addressed. First, it does not explain why we focus on the effects of immigration and trade on discriminatory behaviors observed. Globalization can have various dimensions such as trade, foreign direct investment, immigration, and capital market liberalization. Given the various aspects of globalization, I am wondering why the authors pay particular attention to the two dimensions of globalization – trade and immigration, not trade and FDI or trade capital market liberalization, etc. The authors would justify this point well in the main text.

Response: It is well noted that the impact of globalization is multifaceted, and trade and immigration are only part of the big picture. We now note, in the 2nd sentence in the 4th paragraph in introduction, that “Openness to trade and immigration are important components of the globalized world, and we choose to focus on these two aspects to provide a particular perspective regarding the general relationship between globalization and culture.”

Second, I am wondering how trade and immigration influence discriminatory behaviors observed among ethnically Chinese residents living abroad. I think that a large body of literature in international political economy has explained underlying mechanisms linking trade and immigration to xenophobic behaviors. Yet the authors do not present these mechanisms in a systematic way, while just describing some selective studies supporting their theoretical frameworks. In doing so, the authors do not describe whether trade and immigration have a negative, positive, or null impact on xenophobic behavior observed. The authors would elaborate some testable hypotheses on the effect of trade and immigration on xenophobic behavior observed.

Response: Our reading of the literature regarding trade and immigration suggests that the field is mixed with theories and evidence pointing on both directions. We are not aware of any theory or evidence that directly studies the question we investigate in this project. And we try to cite studies that are most closely related to our specific research question in this context. As a result, we do not have a strong preference toward any particular theory in this context, nor do we have a prior study against which to compare closely. Therefore, we did not set out to seek evidence to validate or invalidate any particular theory. We deliberately try not to frame a specific testable hypothesis that predicts a particular relationship. However, if there are critical omissions on our part regarding the literature review, please kindly let us know, and we are more than happy to address them.

2. Empirics

The authors predict that an increase in trade and immigration can have a negative, positive, or null effect on xenophobic behaviors observed. The questionnaire is “Since the virus outbreak in China, have you noticed any related discriminatory behavior in your working environment and daily life?.” The underlying logic behind the hypothesis is that openness to trade and immigration facilitates more interactions with people outside one’s race, ethnicity and nationality, thus potentially improves mutual understanding and cultural exchanges and mitigating xenophobic sentiments. Given this logic, how can we know that trade and immigration affect local natives in a way that the authors expect? To figure out the mechanisms, wouldn’t it be better to do some survey experiments to local natives rather than ethnically Chinese residents? Simply asking ethnically Chinese residents living abroad does not show that they identify the causal mechanisms between trade and immigration and xenophobic behaviors as local natives are main actors under their theoretical framework.

Response: Our view is that openness to trade and migration may have either a positive or a negative effect. “Openness to trade and migration facilitates more interactions with people outside one's race, ethnicity and nationality, thus potentially improves mutual understanding and cultural exchanges, mitigating xenophobic sentiments”, as we noted in the 4th paragraph of the article, is our synthesis of the views argued in some cited literature (McLaren, 2003; Ellison et al., 2011; Bove and Elia, 2017). It is a plausible mechanism through which openness can mitigate discrimination. But it is not our hypothesis or prediction on the empirical results. To further clarify this point, we now note, in the 3rd sentence of the 4th paragraph, that “They (openness to trade and migration) may mitigate or intensify xenophobic sentiments, and it is an important empirical question to explore the evidence.”

Related to the above point, another concern is that the authors use trade and immigration indicators at the national level. Employing the national-level indicators capturing trade and immigration does not tell that individuals are affected by them. Although countries may be more economically opened in terms of trade and immigration, it does not necessarily mean that individuals know, experience, and perceive them objectively and subjectively. If the authors attempt to uncover the causal mechanisms, it would be better to use some survey questionnaires to measure how individual respondents know, experience, and perceive the degree of trade and immigration at the individual level.

Response: Indeed, the respondents do not necessarily know or experience the openness to trade and migration directly. But it is not what our survey set out to measure, nor is it necessary for our empirical model. The survey was designed to measure the respondent’s experience and perception regarding COVID-related discriminatory behavior. Our empirical model was used to estimate and test whether respondents living in countries with a greater degree of openness systematically experience more or less discrimination. We believe that our empirical approach remains valid as long as the constructed instrumental variables based on country-pairwise geographical relationships are independent of the respondents’ observable characteristics. 

 

Reviewer 2: 

This paper traces the country-specific factors that contribute to anti-Chinese discrimination in other countries under the shadow of a pandemic first spread out in China. There are many merits in this paper. Methodologically, this study is carefully done following Ortega and Peri (2014)'s IV approach using the dyadic bilateral geographical and cultural distance. The topic is very timely and critical as the pandemic has been prevailing around the world. Nonetheless, I have several points of reservation and concern regarding this research.

1. It is entirely in a black box how the respondents are recruited. Even a snowballing sampling, the readers need to know how the respondents are sampled, and who they are, how the survey was done in which language. For instance, it might be possible that these respondents were recruited from the more concerned or more nationalistic population of Chinese immigrants. The platform used in recruitment (WeChat) makes this more likely. Also, it appears that there is no compensation for the survey to the respondents. If you are not paid, what would have been the motivation for the survey-takers to participate in this survey, other than they are particularly concerned about the anti-Chinese atmosphere or feel patriotic about the difficulties their home country was going through? I do not think the WVS analysis remedies this issue.

Response: The respondents are not compensated for taking the survey. Indeed, it is certainly possible that our sampling approach may recruit respondents that are systematically more or less nationalistic. If this potential bias from sample selection is independent across countries, then the OLS estimates would be valid. Otherwise, if the bias is not independent across countries, then it would invalidate the OLS estimates. However, as long as such biases are independent of the instrumental variables we constructed, then our IV estimates remain valid. Because our instruments are constructed using country-pairwise geographical information, we think it is unlikely that the sample selection is correlated with our instruments. In particular, in this revision, we added an additional measure of openness based on shares of trade and migration excluding those from China. We think it is highly plausible that the instruments constructed by leaving China out would be independent of the respondents’ observable characteristics. 

We now note, in the 2nd and 3rd sentence in the 2nd paragraph of page 5, that “For instance, our sampling approach may attract more nationalistic respondents or those with a stronger personal tie to China. If such characteristics are correlated with openness to trade or migration or their residing country, then our OLS estimates would be biased.” The discussion in the following paragraph regarding our IV estimation strategy then serve to address this concern.

2. Conceptually, I was not entirely clear whether the paper is about anti-immigrants, anti-China, anti-Chinese, or anti-Asian, or just broad xenophobia: all these have different implications for hypothesizing and analyses. The timing of the survey was 7 days from Feb 11th to Feb 17th, 2020, which the authors described it was a period when the infection was mostly taking place within mainland China. However, the epidemic already took place in South Korea and Japan (in the cruise ship) on a massive scale and in Taiwan and Hong Kong as well. I think this makes Anti-Asian sentiment a better angle, but at least I hope this issue can be discussed and clarified at the beginning.

Response: The point is well noted. We now revise the 1st sentence in the 4th paragraph of the introduction to read “We are particularly interested in the effect of a country's openness to trade and migration on the likelihood of an ethnically Chinese resident observing anti-Asian discriminatory behaviors.”

3. Empirically, the primary issues I had were related to how to adopt Ortega and Peri's approach. I think this paper's setup is rather China- or East Asia- specific and different from the general perspective in Ortega and Peri. So the right approach would be to take the share of trade with "China" and the share of "Chinese immigrants" in the population, rather than general trade or immigration. Many European countries probably have many immigrants from neighboring European states, but a few from China. I could not think of why and how this would matter in the same way as, say, in South Korea, where a large proportion of immigrants must be from China.

Response: The point is also well taken and very much appreciated. We now expand the main analysis by further decomposing the openness measures into trade and migration with China and those other than China. The analysis lends much richer insights into the paper (results presented in Table 2, and discussed in the results section). We acknowledge your suggestion in footnote 11 on page 6.

4. Second, I am very concerned about the correlation between trade and immigration and the fact that the authors use these variables together in all models. I suspect this might drive the results of the paper. First, the authors use the same IV for both trade and immigration. Second, as shown in Figure 1, the vast majority of responses came from the US, Australia, and Canada: all of them are high immigration and high trade countries. Third, Ortega and Peri (2014) use the two variables separately and together, which I believe this paper should do. Also, the authors need to report the first stage of 2SLS, at least in the appendix.

Response: The point is also well taken and very much appreciated. We expand the main analysis by analyzing openness to trade and migration separately. The results are consistent with findings of the model with both openness measures, and reported in columns (5) and (6) of Table 2. The first-stage estimates on the instruments are also reported now in Table 3, which shows statistically significant relationship between the constructed instruments and the endogenous regressors. 

5. Finally, because all survey was taken after the outbreak of the epidemic and the question was specifically about "since the outbreak," I was not sure if the discrimination got worse than or the same as before. Especially, due the deteriorating relationship between China and the US along with some western countries since at least 2018, maybe the discrimination was rising even before the pandemic.

Response: It is a possible that the perceived discriminatory behavior by the respondents was in fact starting before the COVID crisis. Nonetheless, we think the follow up question regarding the type of COVID-related discrimination is very specifically pointed at COVID, and the results based on those measures (Table 4) are largely consistent with our main results. 

6. I have questions about the data sources: Why trade data (2012-2016) is from CoW, not the World Bank or the WTO? Why are migration data from 1991-2000? There was almost no migration from China to Africa back then?

Response: In the choice of trade and immigration data, we followed Bradford and Chilton (2019, J. Law and Econ) that adopts the Ortega and Peri (2014) framework. However, the comment is well noted that alternative data sources, especially more recent immigration data would improve the credibility of the results. To address this issue, we explored alternative data sources on both trade and immigration. 

On bilateral trade data flows, we were not able to obtain the UN dataset. The WTO bilateral trade data only contains import, but not exports. Our trade measure requires the sum of both in the share of the total GDP. As an alternative, we obtained the IMF bilateral trade data and re-constructed and instruments and re-calculated the estimates. In a nutshell, the results are largely consistent with those obtained based on the CoW trade data, but the instruments based on this dataset is too weak. We report this set of results at the end of the robustness check section.

On bilateral immigration flows, we obtained a more recent dataset (Abel and Cohen, 2019, Scientific Data) which contains immigration flows since 2000 and up to 2015. We now use this dataset as the primary immigration data, and all results are updated. The results based on the original immigration dataset (World Bank) is now relegated to the robustness check section. 

7. Some minor points:

a. P.6. says "approximately 76% of respondents chose one of three categories: (i) racially discriminatory message against Chinese in the media (29%); (ii) racist rhetoric by native residents against Chinese in public (23%); and (iii) shunning (23%)." 76% means these categories are exclusive to each other? What if one respondent experienced many of these?

Response: Yes, the question only allowed for one answer from each respondent. In hindsight, we wish it was set to allow for multiple selections. But given the responses, we make the assumption that the respondent selected the most prominent type of discrimination they experienced. We noted this in footnote 5.

b. On P.4 the authors provide some key descriptive statistics, which is very confusing. "In particular, our preferred estimates show that a respondent residing in a country ranked at the 75th percentile of trade share in GDP in our sample sees a 43% decrease in the likelihood of receiving COVID-19 induced discrimination compared to those in a country ranked at the 25th percentile. In contrast, a respondent in the 75th-percentile country in the share of immigrants in population, on average, has an 86% greater likelihood of receiving COVID-19 induced discrimination compared to those in the counterpart ranked at the 25th percentile." It was unclear because I could not figure out whether the percentile was ascending or descending order. It was only evident after I saw the empirical results in later pages.

Response: Point well taken. In the 3rd paragraph on page 2, where we first mention the 75-25 percentile comparison, we revised the sentences to read “Our results show that greater openness to trade increases the likelihood of reported xenophobic behaviors. In particular, consider two otherwise similar countries, with one country more open to trade (ranked at the 75th percentile in trade share) than the other (ranked at the 25th percentile in trade share). Our preferred estimates show that a respondent residing in the more open country sees an 80% increase in the likelihood of receiving COVID-19 induced discrimination.”

c. Isn't it also possible that observed discrimination can be subject to the fear that the respondents at the time?

Response: It is indeed possible that fear renders respondents more sensitive to discriminatory behavior. But we think it is unlikely that the underlying level of fear would be correlated with our constructed instruments. Therefore, we are confident in our reported estimates. 

d. Table1: how about age? Why public transportation?

Response: We do not report age in the descriptive statics table because the survey only collects a categorical variable on age. The median of our sample lies in the age group of 31-40 year old. We now note this in the text, in the first paragraph under the data section that “The respondents are 65% female, 55% received their highest degree outside China, 32% routinely use public transportation, 65% below the age of 40 and an median age in the 31-40 year old group.” 

Public transportation is a proxy to measure the respondent’s intensity of daily contact with strangers and their residential setting, which may be correlated with the level of discriminatory behavior they are exposed to.

e. Local media on P. 10 mean domestic media?

Response: Local media meant media in the respondent’s residing country. We avoided using the domestic media to avoid the confusion that it might be referring to Chinese domestic media. This set of results are now removed from the revision.

f. Can the authors provide full survey questions and an screenshot of the survey in the appendix?

Response: Yes. The survey questionnaire (a pdf file) is provided with this revision as supplementary.

---

## [Decision Letter · Decision Letter 1]

22 Mar 2021

Openness and COVID-19 induced xenophobia: 

The roles of trade and migration in sustainable development

PONE-D-20-32641R1

Dear Dr. He,

We’re pleased to inform you that your manuscript has been judged scientifically suitable for publication and will be formally accepted for publication once it meets all outstanding technical requirements.

Kind regards,

Shang E. Ha, Ph.D.

Academic Editor

PLOS ONE

Additional Editor Comments (optional):

Reviewers' comments:

Reviewer's Responses to Questions

**Comments to the Author**

1. If the authors have adequately addressed your comments raised in a previous round of review and you feel that this manuscript is now acceptable for publication, you may indicate that here to bypass the “Comments to the Author” section, enter your conflict of interest statement in the “Confidential to Editor” section, and submit your "Accept" recommendation.

Reviewer #1: All comments have been addressed

2. Is the manuscript technically sound, and do the data support the conclusions?

Reviewer #1: Yes

3. Has the statistical analysis been performed appropriately and rigorously? 

Reviewer #1: Yes

4. Have the authors made all data underlying the findings in their manuscript fully available?

Reviewer #1: (No Response)

5. Is the manuscript presented in an intelligible fashion and written in standard English?

Reviewer #1: Yes

6. Review Comments to the Author

Reviewer #1: The authors have done a good job of responding to my concerns and addressing them. I recommend this article for publication.

7. PLOS authors have the option to publish the peer review history of their article (what does this mean?). If published, this will include your full peer review and any attached files.

Reviewer #1: No

---

## [Editor Report · Acceptance letter]

29 Mar 2021

PONE-D-20-32641R1 

Openness and COVID-19 induced xenophobia: The roles of trade and migration in sustainable development 

Dear Dr. He:

I'm pleased to inform you that your manuscript has been deemed suitable for publication in PLOS ONE. Congratulations! Your manuscript is now with our production department. 

Kind regards, 

on behalf of

Dr. Shang E. Ha 

Academic Editor

PLOS ONE